# Differential Modulation of Catecholamine and Adipokine Secretion by the Short Chain Fatty Acid Receptor FFAR3 and α_2_-Adrenergic Receptors in PC12 Cells

**DOI:** 10.3390/ijms25105227

**Published:** 2024-05-11

**Authors:** Deepika Nagliya, Teresa Baggio Lopez, Giselle Del Calvo, Renee A. Stoicovy, Jordana I. Borges, Malka S. Suster, Anastasios Lymperopoulos

**Affiliations:** Laboratory for the Study of Neurohormonal Control of the Circulation, Department of Pharmaceutical Sciences (Pharmacology), Barry and Judy Silverman College of Pharmacy, Nova Southeastern University, Fort Lauderdale, FL 33328, USA; dn635@mynsu.nova.edu (D.N.); tb1985@mynsu.nova.edu (T.B.L.); gd849@mynsu.nova.edu (G.D.C.); rs2981@mynsu.nova.edu (R.A.S.); jb3837@mynsu.nova.edu (J.I.B.); ms5019@mynsu.nova.edu (M.S.S.)

**Keywords:** adrenal chromaffin cell, α_2_-adrenergic receptor, adipokine, beta-hydroxybutyrate, catecholamine, free fatty acid receptor-3, G protein-coupled receptor kinase-2, short chain free fatty acid, signal transduction

## Abstract

Sympathetic nervous system (SNS) hyperactivity is mediated by elevated catecholamine (CA) secretion from the adrenal medulla, as well as enhanced norepinephrine (NE) release from peripheral sympathetic nerve terminals. Adrenal CA production from chromaffin cells is tightly regulated by sympatho-inhibitory α_2_-adrenergic (auto)receptors (ARs), which inhibit both epinephrine (Epi) and NE secretion via coupling to Gi/o proteins. α_2_-AR function is, in turn, regulated by G protein-coupled receptor (GPCR)-kinases (GRKs), especially GRK2, which phosphorylate and desensitize them, i.e., uncouple them from G proteins. On the other hand, the short-chain free fatty acid (SCFA) receptor (FFAR)-3, also known as GPR41, promotes NE release from sympathetic neurons via the Gi/o-derived free Gβγ-activated phospholipase C (PLC)-β/Ca^2+^ signaling pathway. However, whether it exerts a similar effect in adrenal chromaffin cells is not known at present. In the present study, we examined the interplay of the sympatho-inhibitory α_2A_-AR and the sympatho-stimulatory FFAR3 in the regulation of CA secretion from rat adrenal chromaffin (pheochromocytoma) PC12 cells. We show that FFAR3 promotes CA secretion, similarly to what GRK2-dependent α_2A_-AR desensitization does. In addition, FFAR3 activation enhances the effect of the physiologic stimulus (acetylcholine) on CA secretion. Importantly, GRK2 blockade to restore α_2A_-AR function or the ketone body beta-hydroxybutyrate (BHB or 3-hydroxybutyrate), via FFAR3 antagonism, partially suppress CA production, when applied individually. When combined, however, CA secretion from PC12 cells is profoundly suppressed. Finally, propionate-activated FFAR3 induces leptin and adiponectin secretion from PC12 cells, two important adipokines known to be involved in tissue inflammation, and this effect of FFAR3 is fully blocked by the ketone BHB. In conclusion, SCFAs can promote CA and adipokine secretion from adrenal chromaffin cells via FFAR3 activation, but the metabolite/ketone body BHB can effectively inhibit this action.

## 1. Introduction

α_2_-adrenergic receptor (α_2_AR) agonists are important drugs widely used in perioperative patients for sedation and analgesia [1]. α_2_ARs consist of three different subtypes in humans (α_2A_, α_2B_, α_2C_), all of which are G protein-coupled receptors (GPCRs) that couple mainly to Gi/o proteins to inhibit adenylyl cyclase (AC) and cyclic 3′,5′-adenosine monophosphate (cAMP) synthesis in cells [2,3,4]. They play pivotal roles in the presynaptic autoinhibition of norepinephrine (NE) release from sympathetic neurons, as well as of NE and epinephrine (Epi) secretion from the chromaffin cells of the adrenal medulla [5,6,7,8]. These two catecholamines (CAs) are significantly elevated in chronic human heart failure (HF), mediating the sympathetic nervous system (SNS) hyperactivity that results in higher cardiotoxicity for the failing heart [9,10]. Like many other GPCRs, α_2_ARs are subject to GPCR-kinase (GRK)-dependent phosphorylation, which leads to their desensitization, i.e., decoupling from G proteins or G protein signaling termination [11]. Indeed, one of the most abundant and ubiquitously expressed GRKs, GRK2, is upregulated in the adrenal medulla during HF, resulting in severe α_2_AR dysfunction and chronically increased CA secretion [7].

Short-chain free fatty acids (SCFAs), such as propionate and butyrate, are important metabolites serving as energy sources for the heart and other organs and tissues. At the same time, they function as signaling hormones, stimulating free fatty acid receptors (FFARs), which are also plasma membrane-residing GPCRs [12,13,14]. One of the four mammalian FFARs, FFAR3 (also known as GPR41), regulates cardiovascular function via effects in peripheral sympathetic neurons, wherein it promotes neuronal firing and NE release [15]. FFAR3 couples to Gi/o proteins, which can activate phospholipase C (PLC)-β_2/3_ and downstream calcium signaling via their free Gβγ subunits [16]. Synapsin-2b is ultimately phosphorylated to induce vesicle fusion and NE release into the SNS synapse [16]. In addition, the stimulatory effect of propionate/FFAR3 in sympathetic neurons is directly regulated by βARs, via Regulator of G protein Signaling (RGS)-4 protein activation [17]. Nevertheless, it is currently not known whether FFAR3 exerts a similar, facilitatory (or other) effect on adrenal CA secretion, as well.

In the present study, we sought to investigate the role of FFAR3 in chromaffin cell CA secretion and synthesis (if any), as well as its potential crosstalk with adrenal α_2_ARs in the regulation of this process. To this end, we used the rat pheochromocytoma cell line PC12, which cells are essentially chromaffin cells derived from an adrenal medulla tumor and express FFAR3 endogenously [18]. However, they lack endogenous ARs [3]. For this reason, we expressed the human α_2A_AR subtype in them via transfection [5,8]. We also examined the effects of the ketone body metabolite β-hydroxybutyrate (BHB or 3-hydroxybutyrate) on CA production, given that BHB is a known FFAR3 antagonist [19,20,21]. We found that FFAR3 agonism with propionate promotes CA secretion and synthesis in PC12 cells, boosting the effect of nicotinic cholinergic receptor activation, but BHB inhibits this FFAR3-dependent effect. Additionally, GRK2 blockade enhances the α_2A_AR-mediated inhibition of CA production, and the simultaneous blockade of both FFAR3 and GRK2 effectively suppresses CA secretion from PC12 cells. Finally, as further documentation of the inhibitory role of BHB in FFAR3 signaling in chromaffin cells, we found that BHB also suppresses SCFA (propionate)-dependent adipokine secretion [22] from PC12 cells.

## 2. Results

### 2.1. FFAR3 Promotes CA Secretion and Synthesis in PC12 Cells

We first examined whether FFAR3 plays a role in adrenal CA secretion, similar to the role it plays in NE release from sympathetic neurons [15,17]. Indeed, we found that acute FFAR3 stimulation with propionate increased the secretion of both Epi (Figure 1A) and NE (Figure 1B) to a significant extent, albeit lesser than that of nicotine-induced CA secretion (Prop bars gave <100% of nicotine-induced responses in Figure 1A,B). Importantly, the ketone body metabolite BHB, although without effect when applied alone, completely prevented the propionate-induced Epi (Figure 1A) and NE (Figure 1B) secretions, suggesting that BHB acts as an FFAR3 antagonist in PC12 cells. We also assessed TH mRNA induction via real-time PCR, as a measure of CA synthesis, given that TH is the enzyme catalyzing the first and rate-limiting step of CA biosynthesis [6]. As shown in Figure 1C, propionic acid induced TH mRNA expression to a significant extent (though again smaller than that of nicotine induction), an effect also abolished by BHB (Figure 1C). BHB alone was without effect (Figure 1C). Thus, propionic acid stimulates both the synthesis and the secretion of CAs from PC12 cells through FFAR3, an effect inhibited by the ketone BHB.

### 2.2. GRK2 Inhibition Enhances a_2A_AR-Dependent Suppression of CA Production in PC12 Cells

We next explored the role of GRK2-mediated phosphorylation and desensitization in the inhibitory function of α_2A_AR in terms of CA secretion. As expected, α_2A_AR activation with brimonidine (UK14304), a full α_2_AR agonist, partially inhibited both Epi (Figure 2A) and NE (Figure 2B) secretion in response to nicotine (i.e., cholinergic stimulation—the physiologic stimulus) in PC12 cells. The acute pharmacological inhibition of GRK2 with Cmpd101 [23] alone has no effect on nicotine-induced CA secretion per se (Figure 2A,B), again as expected, given that nicotine activates nicotinic receptors that are not GPCRs, i.e., GRK2 substrates. Upon concomitant treatment with brimonidine, however, GRK2 blockade markedly boosted the α_2_AR agonist’s sympatholytic effect, almost completely suppressing nicotine-induced Epi (Figure 2A) and NE (Figure 2B) secretion. The same was true for CA synthesis, as well. As shown in Figure 2C, brimonidine partially blocked nicotine-induced TH mRNA upregulation, but, in the presence of GRK2 blockade, α_2A_AR activation led to a more robust inhibition of nicotine-induced TH mRNA upregulation. GRK2 blockade alone had, again, no effect on nicotine-dependent TH upregulation (Figure 2C). Taken together, these findings suggest that GRK2 blockade boosts α_2_AR sympatholytic activity in PC12 cells, even at normal levels of GRK2 expression (i.e., in the absence of GRK2 upregulation).

### 2.3. Suppression of Propionate-Nicotine Combination’s Effect on CA Secretion Requires Both a_2_AR Activation and FFAR3 Blockade in PC12 Cells

We also examined the interplay between the pro-secretory FFAR3 and the anti-secretory α_2A_AR in relation to cholinergic-dependent CA secretion in PC12 cells. Co-administration with propionic acid more than doubled the Epi (Figure 3A) and NE (Figure 3B) secretion responses of PC12 cells to nicotine, indicating a synergistic effect of FFAR3 activation with cholinergic (vagal) stimulation on the adrenal chromaffin cell CA secretion. Brimonidine could only partially suppress CA secretion in response to either nicotine alone or the combination of nicotine and propionate (Figure 3A for Epi, Figure 3B for NE). Similarly, BHB pretreatment only partially blocked Epi (Figure 3A) and NE (Figure 3B) secretion in response to nicotine and propionate combined, while it had no effect on the response to nicotine alone. In other words, BHB essentially blocked only the additive effect of FFAR3 on nicotine-dependent CA secretion. Importantly, when combined, brimonidine and BHB were able to substantially (>70%) suppress Epi (Figure 3A) and NE (Figure 3B) secretions in response to the nicotine plus propionate combination. Taken together, these data indicate that SCFAs, similar to propionic acid, act synergistically with cholinergic stimulation to markedly enhance adrenal CA secretion. Thus, the addition of an FFAR3 antagonist, like BHB, to an α_2_AR agonist can provide effective sympatholysis in adrenal chromaffin cells.

### 2.4. Suppression of Propionate-Induced Adipokine Secretion by BHB in PC12 Cells

Finally, we tested whether BHB, acting as an FFAR3 antagonist, can also inhibit SCFA-induced adipokine secretion from PC12 cells. Adipokines, such as leptin and adiponectin, are crucial for adipocyte differentiation (adipogenesis), but are also known to be involved in the inflammation of adipose and various other tissue types [24,25,26,27]. SCFAs, specifically butyrate, have been reported to promote adiponectin synthesis and secretion from pre-adipocytes via FFAR3 [22], an effect that mediates butyrate/FFAR3-dependent adipogenesis [22]. Indeed, we found that acute FFAR3 stimulation with propionate increased the secretion of leptin (Figure 4A) and adiponectin (Figure 4B), while BHB, although without effect when applied alone, completely prevented these propionate-induced leptin (Figure 4A) and adiponectin (Figure 4B) secretions from PC12 cells. This finding provides additional evidence for the role of BHB as an FFAR3 antagonist in PC12 cells.

## 3. Discussion

In the present study, we report, for the first time to our knowledge, that FFAR3 and α_2_ARs function in an opposing manner on adrenal CA secretion (Figure 5). Although α_2_ARs inhibit further CA release, as they have been documented extensively to do [6,7,8], the SCFA receptor FFAR3 promotes NE and Epi secretion, as well as synthesis (as reflected by TH mRNA upregulation), from chromaffin cells (Figure 4). Therefore, it appears that FFAR3 is capable of not only enhancing NE release from peripheral sympathetic neurons [15,16], but also of promoting NE and Epi secretion from the adrenal medulla. Importantly, BHB blocks this FFAR3-dependent effect on chromaffin cell CA production, similarly to what GRK2 blockade on the α_2_AR does (Figure 4). If these pathways, uncovered by the present study in PC12 cells, also operate in human adrenal glands in vivo, then ketone bodies like BHB may be of therapeutic value for lowering the excessive SNS activity that accompanies and aggravates chronic human HF [9,10].

FFAR3 is expressed in post-ganglionic sympathetic neurons, including cardiac sympathetic nerve terminals, wherein it promotes neuronal firing and activity resulting in elevated NE release [15]. Although both NE and Epi mediate the effects of the SNS on all cells and tissues, NE is the neurotransmitter synthesized, stored, and released from sympathetic neurons, whereas Epi is a hormone synthesized in the adrenal medulla and secreted into the systemic circulation [4,28]. FFAR3-knockout mice display significantly lower CA synthesis, as evidenced by TH downregulation, the enzyme that catalyzes the rate-limiting step of CA biosynthesis [15], as well as lower sympathetic neuronal firing rate and heart rate [15]. Mechanistically, FFAR3 stimulates NE release via the Gi/o protein-derived free Gβγ subunit activation of PLCβ_2/3_ [15,16]. PLCβ_2/3_, in turn, induces Ca^2+^ signaling, which ultimately results in the activation of extracellular signal-regulated kinase (ERK)1/2, which then phosphorylate synapsin-2b at Ser426 to induce vesicle fusion with the neuronal plasma membrane and subsequent NE release from sympathetic nerve endings [16]. This ability of FFAR3 to promote Ca^2+^-dependent synaptic vesicle fusion and exocytosis via the Gi/o protein-derived free Gβγ subunits probably explains why the α_2_AR inhibits CA secretion, whereas FFAR3 promotes it, despite both receptors coupling to the same G protein type (Gi/o) in adrenal chromaffin cells (Figure 5). In any case, the precise signaling pathways linking both receptors to their effects on CA secretion in PC12 cells await elucidation in future studies.

BHB is a major ketone produced metabolically by FFAs, and it exerts a variety of cardioprotective effects for the failing heart, including direct cardiac effects, such as improved mitochondrial respiration, cardiomyocyte function, and myocardial blood flow, as well as reduced inflammation, fibrosis, oxidative stress, and hypertrophy [29,30,31,32]. Indirect systemic effects such as immune modulation, weight loss, sympatholysis, and reduced vascular resistance also contribute to ketones’ beneficial cardiovascular effects [32]. BHB is also among the ketone bodies whose production is elevated by sodium/glucose co-transporter-2 (SGLT2) inhibitor drugs, such as dapagliflozin and empagliflozin [32,33,34]. These drugs exert a plethora of beneficial effects in the cardiovascular system, beyond their anti-diabetic effects [35,36]. BHB probably mediates a vast portion of SGLT2 inhibitors’ cardiometabolic and anti-inflammatory benefits. Our present study adds one more to this ever-expanding list of beneficial BHB effects: reduction in CA production and secretion from adrenal chromaffin cells, which is clearly an anti-stressogenic mechanism. The mechanism for this sympatholytic effect of BHB is FFAR3 antagonism. BHB has been documented to act as an FFAR3 blocker, although one study claimed the opposite, i.e., that it works as an FFAR3 agonist [37]. In our hands, BHB operated clearly as an FFAR3 antagonist, opposing all the effects of propionic acid in PC12 cells that we tested. Of note, we recently reported that Regulator of G protein signaling (RGS)-4 terminates FFAR3 signaling to produce cardio-protection and reduce cardiac SNS activity [17]. It would therefore be quite interesting to examine the potential crosstalk/interplay between RGS4 and BHB for converging actions on FFAR3 in cardiac, neuronal, and adrenal chromaffin cells, and we are planning to do this in future studies.

In the present study, we additionally reaffirmed the central role of GRK2 in the regulation of the sympatho-inhibitory α_2_AR signaling/function towards the inhibition of CA secretion in chromaffin cells. Importantly, long-term α_2_AR activation itself can upregulate GRK2 levels in chromaffin cells, thereby inducing a negative feedback loop on its signaling, which turns into a positive feedback loop for adrenal CA secretion [8]. Our present results with Cmpd101 recapitulate our old findings with βARKct [4,7], and strongly suggest that GRK2 inhibition can enact a potent sympatholytic action in the adrenal gland, similarly to BHB acting through FFAR3 blockade.

Finally, we also show, for the first time to our knowledge, that FFAR3 activated by SCFAs directly stimulates the production and secretion of certain important adipokines, specifically of leptin and adiponectin, in PC12 cells, while the ketone BHB is capable of fully suppressing this effect, as well. Adipokines are cytokines produced and secreted by adipocytes, and several of them, particularly leptin and adiponectin, are involved in the modulation of obesity-associated inflammation, but also in neuroinflammation, cardiovascular inflammation, atherosclerosis, and inflammation of other tissues and organs [24,25,26,27]. Interestingly, leptin, in particular, is associated with increased SNS activity in humans (as part of its anorexigenic/food intake-reducing effect) [38], and, specifically in chromaffin cells, has been reported to modulate CA secretion. In mouse chromaffin cells, it was shown to increase CA secretion upon sustained stimulation (albeit it reduced action potential firing at rest) [39] and, in PC12 cells, leptin enhanced vesicle trafficking and the secretion of CAs via Ca^2+^-dependent signaling, whereas CAs inhibited leptin secretion through βARs [40]. Therefore, FFAR3-induced leptin secretion in PC12 cells might be another mechanism by which SCFAs elevated CA secretion from the adrenal medulla. Moreover, the BHB-mediated inhibition of FFAR3 signaling towards leptin and adiponectin production in PC12 cells, reported in the present study, may represent another mechanism of the sympatholytic and inflammation-modulating effects of BHB in the adrenal gland. The precise roles of these adipokines in the FFAR3-dependent regulation of CA secretion from adrenal chromaffin cells are certainly worth investigating in future studies.

The present study has two major limitations. One, although they are physiologically relevant (rat adrenal chromaffin) cells, these cells were not primary, bona fide differentiated cells, since PC12 is a cancer (pheochromocytoma) cell line (for instance, PC12 cells lack endogenous ARs, unlike primary adrenal chromaffin cells [41]). The other limitation is that our study was performed exclusively in vitro; our findings obviously require confirmation in in vivo models and settings. Nevertheless, given that the cell line employed closely mimics chromaffin cells and is physiologically relevant, our present findings are quite likely to hold true in adrenal glands in vivo, as well.

## 4. Materials and Methods

### 4.1. Materials

All drugs and chemicals were from Sigma-Aldrich (St. Louis, MO, USA).

### 4.2. Cell Culture and Transfections

PC12 cells were purchased from American Type Culture Collection (ATCC, Manassas, VA, USA) and transfected with the human α_2A_-AR cDNA (Missouri S&T cDNA Resource Center, Rolla, MO, USA) via the Lipofectamine method (Invitrogen, Carlsbad, CA, USA), as previously described [8]. Culturing of PC12 cells was performed under standard protocols, as previously described [5,8]. For α_2_AR B_max_ determination purposes, a separate flask with transfected cells was harvested for saturation ligand binding using the α_2_AR-specific antagonist [^3^H]-RX821002 (specific activity 45–65 Ci/mmol, Perkin Elmer, Waltham, MA, USA), again as previously described [8]. To obtain stably transfected clones, the human α_2A_-AR cDNA was inserted into the pD2500 mammalian stable expression vector (DNA 2.0, Menlo Park, CA, USA) and, following transfection via the Lipofectamine method, the selection of stable transfectants was performed with 500 mg/mL Hygromycin B. Radioligand binding confirmed that the transgene (α_2A_AR) was properly expressed, and a stable clone expressing 500 fmol α_2A_AR/mg of total cellular protein was chosen for further experiments.

### 4.3. CA and Adipokine Secretion Assays

In vitro Epi and NE secretion in response to various treatments was measured in the supernatant by ELISA (Bi-CAT (Epinephrine and Norepinephrine) ELISA, 17-BCTHU-E02.1; ALPCO Diagnostics, Salem, NH, USA), as described previously [7]. Leptin and adiponectin were measured via multiplexed ELISA, with a customized “Mix and Match” Rat Cytokine ELISA Profiling Kit (EA-1001; Signosis, Santa Clara, CA, USA), according to the manufacturer’s instructions [42]. Briefly, 2 h before the treatments, the cells were placed in 500 μL serum-free medium (96-well plate) and the supernatants (media) were collected 20 min post-agent application. Then, 100 μL samples were loaded into the ELISA plate for hormone measurements. Detection was based on biotin-labeled antibodies followed by streptavidin–HRP-conjugated secondary antibody incubation. Optical density was ultimately measured at 450 nm.

### 4.4. Real-Time PCR

Total RNA isolation from PC12 cells, reverse transcription to cDNA and quantitative real-time PCR for the (rat) tyrosine hydroxylase (TH) gene were carried out as previously described [7]. The oligonucleotide primer pairs used were 5′-GAAGGGCCTCTATGCTACCCA-3′ and 5′-TGGGCGCTGGATACGAGA-3′ for (rat) TH (Annealing temperature: 63 °C); and 5′-TCAAGAACGAAAGTCGGAGG-3′ and 5′-GGACATCTAAGGGCATCAC-3′ for 18S rRNA (Annealing temperature: 60 °C). Quantitative real-time PCR was performed using a MyIQ Single-Color Real-Time PCR detection system (Bio-Rad, Hercules, CA, USA) using SYBR Green Supermix (Bio-Rad) and 100 nM of oligos. The quantification of mRNA included normalization to 18s rRNA levels. Specific PCR products were determined using melting curves and gel electrophoresis, as described [7]. No bands were seen in the control reactions (in the absence of reverse transcriptase).

### 4.5. Statistical Analysis

Data are presented as the mean ± S.D. values. Statistical differences were assessed by one-way analysis of variance (ANOVA) (GraphPad Software Inc., La Jolla, CA, USA) followed by the Tukey test for multiple comparisons. *p* values less than 0.05 were considered statistically significant.

## 5. Conclusions

SCFAs, such as propionic acid and butyric acid, can promote CA production from adrenal chromaffin cells, as well as NE release from sympathetic neurons, via FFAR3 activation. BHB, acting as an antagonist of FFAR3, inhibits SCFA-dependent CA secretion, and thus can exert sympatholysis in the adrenal gland, similarly to what adrenal α_2_AR agonists and GRK2 inhibition do. Given that FFAR3 appears to promote acetylcholine’s stimulation of CA secretion, gut microbiota perturbations that result in elevated SCFA production may have the additional adverse effect of SNS activity elevation, which could potentially be treated with ketone bodies such as BHB. Additionally, BHB acts as a suppressor of SCFA-induced adipokine production in chromaffin cells, an effect that may portend additional sympatholytic, as well as anti-inflammatory and anti-adipogenic, properties for this ketone body. Sympatholytic mechanisms such as the one uncovered in the present study may contribute significantly to the purported cardiovascular benefits of ketone bodies and of drugs like the SGLT2 inhibitors, which shift cellular metabolism to ketone body production in vivo.

## Figures and Tables

**Figure 1 ijms-25-05227-f001:**
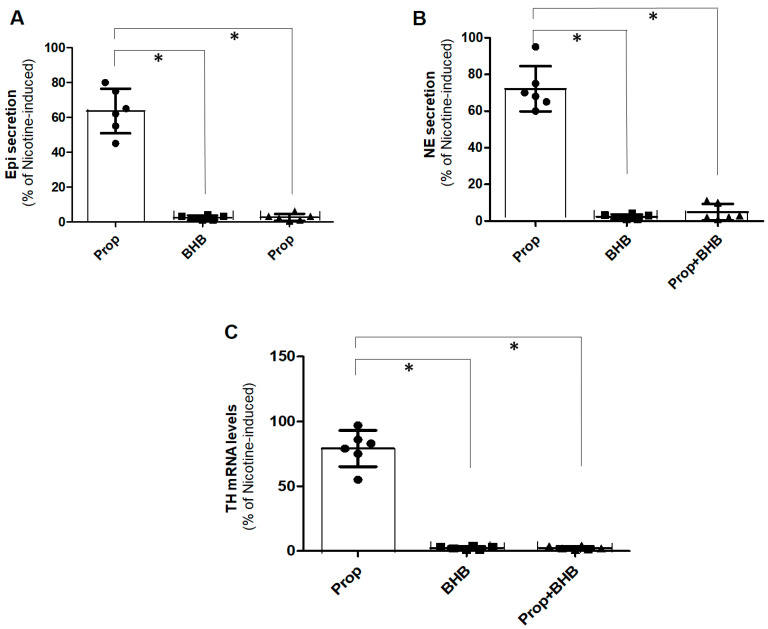
FFAR3 and CA production in PC12 cells. (**A**,**B**) In vitro Epi (**A**) and NE (**B**) secretion from cultured PC12 cells treated with 1 mM propionate (Prop) or 1 mM BHB (or both) for 20 min. Responses are shown as % of the response of the cells to a 50 μM nicotine challenge (nicotine-induced). (**C**) TH mRNA levels in the same cells treated with the same concentrations of the same drugs but after 12 h of treatment. For all panels: *, *p* < 0.05; n = 5 independent experiments per condition.

**Figure 2 ijms-25-05227-f002:**
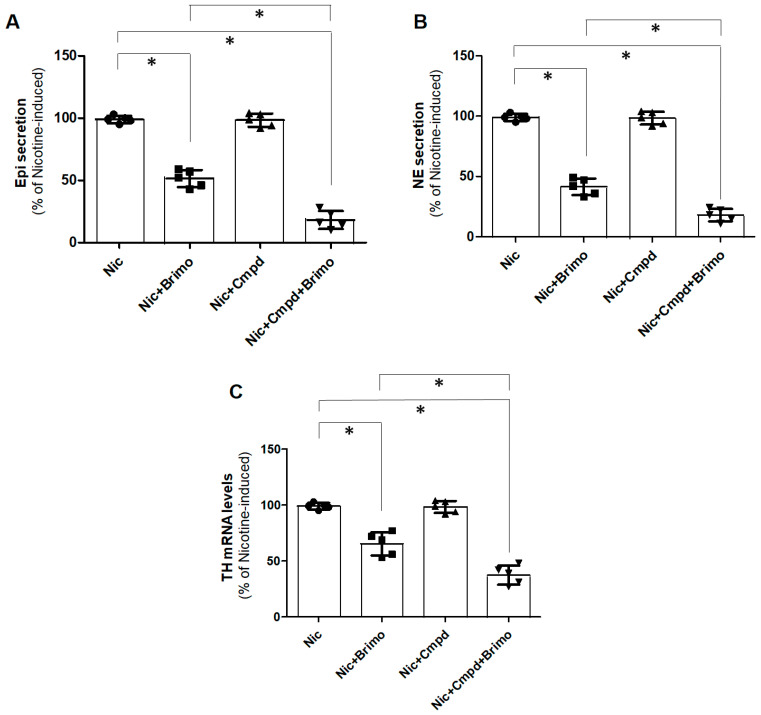
GRK2 and α_2A_AR regulation of CA production in PC12 cells. (**A**,**B**) In vitro Epi (**A**) and NE (**B**) secretion from cultured PC12 cells pretreated with 10 μM brimonidine (Brimo) or 50 μM Cmpd101 (Cmpd) (or both), prior to a 50 μM nicotine (Nic) challenge for 20 min. Responses are shown as % of the response of the cells to a 50 μM nicotine challenge post-vehicle (0.5% DMSO) pretreatment (nicotine-induced, “Nic” alone). (**C**) TH mRNA levels in the same cells treated with the same concentrations of the same drugs but after 12 h of nicotine exposure. For all panels: *, *p* < 0.05; n = 5 independent experiments per condition.

**Figure 3 ijms-25-05227-f003:**
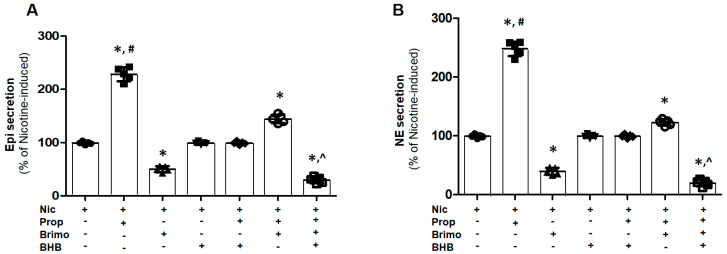
FFAR3 and α_2A_AR crosstalk in the regulation of CA secretion from PC12 cells. In vitro Epi (**A**) and NE (**B**) secretion from cultured PC12 cells pretreated with 1 mM propionic acid (Prop), 10 μM brimonidine (Brimo), 1 mM BHB or combinations thereof, prior to a 50 μM nicotine (Nic) challenge for 20 min. Responses are shown as % of the response of the cells to a 50 μM nicotine challenge alone (i.e., post-vehicle (0.5% DMSO) pretreatment: “nicotine-induced” or “Nic” alone). *, *p* < 0.05; vs. “Nic” alone; ^#^, *p* < 0.05; vs. any other condition; ^^^, *p* < 0.05; vs. “Nic+Brimo” or “Nic+Prop+Brimo” or “Nic+Prop+BHB”; n = 5 independent experiments per condition.

**Figure 4 ijms-25-05227-f004:**
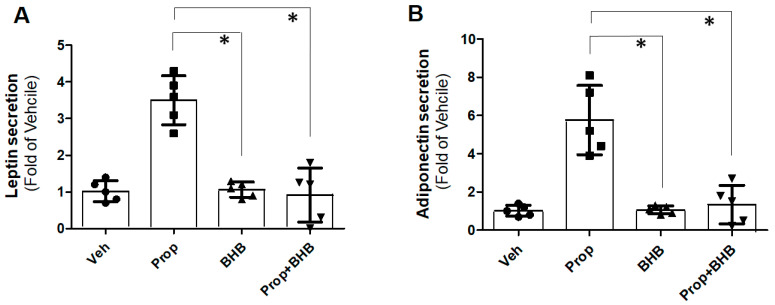
FFAR3 and adipokine production in PC12 cells. In vitro leptin (**A**) and adiponectin (**B**) secretion from cultured PC12 cells treated with 1 mM propionate (Prop) or 1 mM BHB (or both). Responses are shown as fold of the vehicle (Veh, 0.5% DMSO) response. *, *p* < 0.05; n = 5 independent experiments per condition.

**Figure 5 ijms-25-05227-f005:**
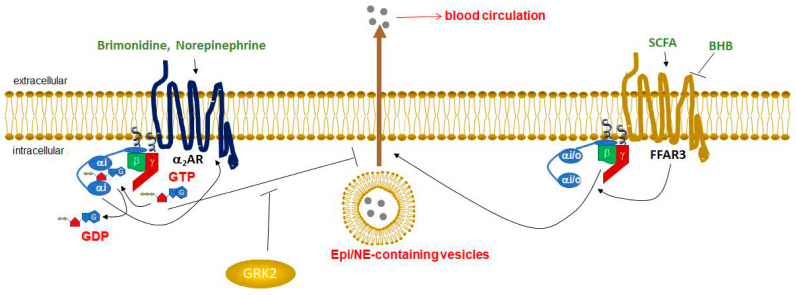
Opposing roles of α_2_AR and FFAR3 in CA secretion and BHB-suppressed FFAR3-dependent adipokine secretion in chromaffin cells. αi: Inhibitory G protein alpha subunit. αi/o: Inhibitory/other G protein alpha subunit. GTP: Guanosine triphosphate. GDP: Guanosine diphosphate. SCFA: Short-chain fatty acid (e.g., propionate). BHB: Beta-hydroxy (or 3-hydroxy) butyric acid. See text for details and for all other molecular acronym descriptions.

## Data Availability

All source data files are available upon request to the correspondence author.

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
