# Peer review of "Differential Modulation of Catecholamine and Adipokine Secretion by the Short Chain Fatty Acid Receptor FFAR3 and α2-Adrenergic Receptors in PC12 Cells"

_ijms, 2024, doi:10.3390/ijms25105227_

Round 1

Reviewer 1 Report

Comments and Suggestions for Authors

The manuscript "Differential modulation of catecholamine and adipokine secretion by the short chain fatty acid receptor FFAR3 and a2-adrenergic receptors in PC12 cells" By Nagliya, et al describes experiments that study the dual regulation of catecholamine and adipokine secretion using PC12 cells and mostly pharmacological tools.

Overall, the experiments are well conducted the results are presented clearly and the limitations of the study are discussed. There few issues that should be addressed.

1.       The study relies on the secretion assays. I would suggest adding more information on how the assay is done.

2.       In relation to point 1, please supply examples of raw data. It is difficult to evaluate the data from the normalized values presented in all the figures.

3.       In addition, I think it will be helpful to show individual data points for the bar graphs, and show SD instead of SEM.

4.       Figure 1 - In line 93 it is stated that the results suggest that BHB acts as a FFAR3 antagonist in PC12 chromaffin cells. That is somewhat confusing - the results show that BHB has an effect, but its target is not known. Is it possible to attribute this effect to the FFAR?

5.       Figure 2 - Can the authors verify the expression of the alpha 2 adrenergic and the level of expression? I think blocking the receptor by antagonist and/or by PTX, for example, will make the case clearer.

6.       Cmpd101 is both GRK2 and GRK3 blocker. How can the authors ascertain it is GRK2?

7.       It was shown that other GPCRs (e.g. muscarinic receptors Inoue M, Matsuoka H, Harada K, Kao LS. Muscarinic receptors in adrenal chromaffin cells: physiological role and regulation of ion channels. Pflugers Arch. 2018 Jan;470(1):29-38) also regulate secretion. Can the authors rule out the possibility that Cmpd101 does not affect other receptors signaling?

8.       Line 139 - synergistic effect of FFAR3 activation with cholinergic (vagal) stimulation on adrenal chromaffin cell is suggested. It is not clear to me what is the basis for this suggested. Is it possible that these are additive effects? A clarification is needed here.

Comments on the Quality of English Language

Some minor editing is needed

Author Response

The manuscript "Differential modulation of catecholamine and adipokine secretion by the short chain fatty acid receptor FFAR3 and a2-adrenergic receptors in PC12 cells" By Nagliya, et al describes experiments that study the dual regulation of catecholamine and adipokine secretion using PC12 cells and mostly pharmacological tools.

Overall, the experiments are well conducted the results are presented clearly and the limitations of the study are discussed. There few issues that should be addressed.

  1. The study relies on the secretion assays. I would suggest adding more information on how the assay is done.

Author response: We thank this reviewer for the overall kind and positive comments about the quality of our work. The reviewer makes a fair point here. We have added more information in section 4.3 of the revised manuscript (lines 296-301, highlighted in yellow).

  1. In relation to point 1, please supply examples of raw data. It is difficult to evaluate the data from the normalized values presented in all the figures.

Author response: Apologies but we are not sure what the reviewer means here by examples of “raw data”. All data presented are comparisons among the various conditions; thus, showing raw data makes no sense and, in fact, might confuse the reader. We hope the reviewer understands.

  1. In addition, I think it will be helpful to show individual data points for the bar graphs, and show SD instead of SEM.

Author response: Done, thank you for the suggestion.

  1. Figure 1 - In line 93 it is stated that the results suggest that BHB acts as a FFAR3 antagonist in PC12 chromaffin cells. That is somewhat confusing - the results show that BHB has an effect, but its target is not known. Is it possible to attribute this effect to the FFAR?

Author response: We thank this reviewer for this valid point. It is true that BHB can have additional effects more than simply blocking FFAR3. However, according to the current literature (Refs. 19-21) and based on our present findings (direct antagonism of propionate`s effects), BHB clearly acts as an FFAR3 antagonist, at least in PC12 cells.

  1. Figure 2 - Can the authors verify the expression of the alpha 2 adrenergic and the level of expression? I think blocking the receptor by antagonist and/or by PTX, for example, will make the case clearer.

Author response: We have documented the expression of α2A-AR in transfected PC12 cells in previous publications (Refs. 2 & 8) and describe the generation and characteristics of these cells in detail in section 4.2, including the receptor expression levels (Bmax: ~500 fmol per mg of protein).

  1. Cmpd101 is both GRK2 and GRK3 blocker. How can the authors ascertain it is GRK2?

Author response: The reviewer makes another excellent point here. It is true that GRK3 is also blocked by Cmpd101 and, unfortunately, there is no agent, to our knowledge, currently certified to sufficiently discriminate between GRK2 and GRK3. However, we have no evidence that GRK3 is involved in regulation of α2AR-dependent CA secretion inhibition, nor has the presence of GRK3 in PC12 cells ever been reported in the literature.

  1. It was shown that other GPCRs (e.g. muscarinic receptors Inoue M, Matsuoka H, Harada K, Kao LS. Muscarinic receptors in adrenal chromaffin cells: physiological role and regulation of ion channels. Pflugers Arch. 2018 Jan;470(1):29-38) also regulate secretion. Can the authors rule out the possibility that Cmpd101 does not affect other receptors signaling?

Author response: We thank this reviewer for another observant comment. Given that GRK2 can desensitize muscarinic receptors, we cannot rule out this possibility. Nevertheless, Cmpd101 was used in the present study in conjunction with brimonidine and nicotine only, i.e., we only measured Cmpd101`s effect on α2AR function and on nicotinic receptor signaling, respectively. Nicotine does not activate muscarinic receptors but only nicotinic receptors, which, as non-GPCRs, are not GRK2/3 substrates (as confirmed also by our data in Fig. 2 showing no effect of Cmpd101 on nicotine-induced CA secretion).

  1. Line 139 - synergistic effect of FFAR3 activation with cholinergic (vagal) stimulation on adrenal chromaffin cell is suggested. It is not clear to me what is the basis for this suggested. Is it possible that these are additive effects? A clarification is needed here.

Author response: Our data presented in Fig. 3 suggest that propionate more than doubles nicotine`s effect on CA secretion, which, by definition, is “synergism”. However, we agree with the reviewer that the effects of FFAR3 activation would probably be better characterized as additive, rather than synergistic, so we have now refrained from using the term “synergistic” in the revised text. We hope now this satisfies this reviewer.

Reviewer 2 Report

Comments and Suggestions for Authors

Comments on manuscript IJMS-2954024

entitled

“Differential modulation of catecholamine and adipokine secretion by the short chain fatty acid receptor FFAR3 and alpha2-adrenergic receptors in PC12 cells” by D. Nagliya et al..

This manuscript describes studies on the release of catecholamines (CA) and the expression of tyrosine hydroxylase, which were conducted exclusively in the rat PC12 cell line. The authors investigated the effects of various compounds on the release of the CAs epinephrine (Epi) and norepinephrine (NE): propionate and BHB as endogenous ligand and antagonist, respectively, at FFAR3, and brimonidine and compund101 as modulators of alpha2-adrenergic receptor (a2-AR) signaling. Experiments were done in nicotine treated cells or in untreated cells. 

In addition to CA release, the effects of FFAR3 modulation on leptin and adiponectin secretion were measured. The reason for that is not really made clear – especially given the fact that the adrenal synthesis of these hormones is more or less negligible in comparison to the synthesis in fat tissue. This makes it look like a filler for the results section.

Sadly, the whole study relies on compounds, which were used at only a single concentration each, and on two analytical techniques, ELISA and qPCR. Thus, all conclusions as to the underlying mechanisms of the observed effects are by inference only.

Major points:

1.     The cell system used is quite artificial: the PC12 cell line had to be transfected with the a2A-AR, to correct at least this deviation from real chromaffin cells. Thus, it is not warranted to put them on par with real chromaffin cells. The term “chromaffin-like” would better fit throughout the manuscript (from abstract to conclusions) - to the reviewer’s knowledge PC12 cells have to treated with glucocorticoids to make them chromaffin-like.

2.     At least one experimental aspect should be shown in real chromaffin cells to show that the findings are really applicable to them. Obviously, the authors are aware of that fact as the identify it as limitation in the discussion.

3.     The mechanistic conclusions should be backed by knock-out or knock-down of FFAR3.

4.     Referencing is quite well, but apparently includes an impressive percentage of self-cites, some of which are not the optimal ref at the respective position and should be replaced by the original paper proving the respective fact (the most extreme example is ref 42, as the authors have stated that they followed manufacturer’s instructions – this paper neither contains the term leptin nor adiponectin). Ref 18 is misleading as it doesn’t prove that PC12 cells resemble real chromaffin cells. Ref 6 should be replaced by the original paper showing the importance of TH in CA biosynthesis.

Minor points:

-       Line 104: Typo, should read µM

-       Lines 200-204: font size is oscillating….

-       Fig 5: In case leptin and adiponectin should be mentioned at all, it would be more appropriate to indicate secretion for them (bring them to other side of membrane…

-       Line 296: Typo, should read ALPCO

Author Response

This manuscript describes studies on the release of catecholamines (CA) and the expression of tyrosine hydroxylase, which were conducted exclusively in the rat PC12 cell line. The authors investigated the effects of various compounds on the release of the CAs epinephrine (Epi) and norepinephrine (NE): propionate and BHB as endogenous ligand and antagonist, respectively, at FFAR3, and brimonidine and compund101 as modulators of alpha2-adrenergic receptor (a2-AR) signaling. Experiments were done in nicotine treated cells or in untreated cells. 

In addition to CA release, the effects of FFAR3 modulation on leptin and adiponectin secretion were measured. The reason for that is not really made clear – especially given the fact that the adrenal synthesis of these hormones is more or less negligible in comparison to the synthesis in fat tissue. This makes it look like a filler for the results section.

Sadly, the whole study relies on compounds, which were used at only a single concentration each, and on two analytical techniques, ELISA and qPCR. Thus, all conclusions as to the underlying mechanisms of the observed effects are by inference only.

Major points:

  1. The cell system used is quite artificial: the PC12 cell line had to be transfected with the a2A-AR, to correct at least this deviation from real chromaffin cells. Thus, it is not warranted to put them on par with real chromaffin cells. The term “chromaffin-like” would better fit throughout the manuscript (from abstract to conclusions) - to the reviewer’s knowledge PC12 cells have to treated with glucocorticoids to make them chromaffin-like.

Author response: We thank this reviewer for the overall kind and positive comments about the quality of our work. Although we disagree that PC12 cells are not chromaffin cells, since they are derived from a chromaffin cell tumour (they are just not normal chromaffin cells), we have refrained from using “chromaffin” and “PC12” next to each other in the revised manuscript. We hope this now satisfies this reviewer.

  1. At least one experimental aspect should be shown in real chromaffin cells to show that the findings are really applicable to them. Obviously, the authors are aware of that fact as the identify it as limitation in the discussion.

Author response: Apologies but we do not understand this comment. PC12 cells are real chromaffin cells, because they perform the most characteristic function of a chromaffin cell: catecholamine synthesis and secretion! They are just not normal chromaffin cells, since they are immortalized (cancerous). This is probably the reason why they do not express any adrenergic receptors, unlike primary chromaffin cells that do. In any case, we have acknowledged this as a limitation in our “Discussion”.

  1. The mechanistic conclusions should be backed by knock-out or knock-down of FFAR3.

Author response: We thank the reviewer for this valid point. Unfortunately, we don`t have the time to perform these experiments given that the journal gave us only 10 days to revise our manuscript. In any case, we are quite confident about the role of FFAR3 we describe, since propionate is a well-known agonist, and BHB has been documented as an antagonist of this receptor.

  1. Referencing is quite well, but apparently includes an impressive percentage of self-cites, some of which are not the optimal ref at the respective position and should be replaced by the original paper proving the respective fact (the most extreme example is ref 42, as the authors have stated that they followed manufacturer’s instructions – this paper neither contains the term leptin nor adiponectin). Ref 18 is misleading as it doesn’t prove that PC12 cells resemble real chromaffin cells. Ref 6 should be replaced by the original paper showing the importance of TH in CA biosynthesis.

Author response: We used Ref. 18 as a previously published study showing FFAR3 is expressed in PC12 cells, not as proof that PC12 cells are chromaffin cells. As for Ref. 42, we used it as reference for the ELISA kit and methodology used in the present study. Ref. 42 did not measure leptin or adiponectin but the ELISA procedure it describes is identical to the one we used in this study. Nevertheless, we have replaced Ref. 42 with a more relevant one in the revised manuscript (highlighted in yellow). We hope this now satisfies this reviewer.

Minor points:

-       Line 104: Typo, should read µM

Author response: Done.

-       Lines 200-204: font size is oscillating….

Author response: Done.

-       Fig 5: In case leptin and adiponectin should be mentioned at all, it would be more appropriate to indicate secretion for them (bring them to other side of membrane…

Author response: We are not sure but we think the reviewer means here to show leptin/adiponectin on the extracellular side of the membrane. We showed them on the intracellular side because they probably need to be synthesized inside the cell first before being secreted. In any case, we agree with the reviewer that these two do not need to be shown in the figure, so we have now removed them from revised Fig. 5.

-       Line 296: Typo, should read ALPCO

Author response: Done.

Round 2

Reviewer 1 Report

Comments and Suggestions for Authors

Thank you for addressing the comments.